# Population preferences for AI system features across eight different decision-making contexts

Søren Holm[1,2]* , Thomas Ploug[3]

**1** Centre for Social Ethics and Policy, School of Law, University of Manchester, Manchester, United Kingdom, **2** Faculty of Medicine, Centre for Medical Ethics, HELSAM, University of Oslo, Oslo, Norway, **3** Department of Communication and Psychology, Centre of Applied Ethics and Philosophy of Science, Aalborg University, Copenhagen, Denmark

☯ These authors contributed equally to this work.
* soren.holm@manchester.ac.uk

**Data Availability Statement:** All data are available at Figshare DOI 10.48420/23309537.

**Funding:** TP 2027-00140B Independent Research Fund Denmark dff.dk The funders had no role in

## Abstract

Artificial intelligence systems based on deep learning architectures are being investigated as decision-support systems for human decision-makers across a wide range of decision-making contexts. It is known from the literature on AI in medicine that patients and the public hold relatively strong preferences in relation to desirable features of AI systems and their implementation, e.g. in relation to explainability and accuracy, and in relation to the role of the human decision-maker in the decision chain. The features that are preferred can be seen as 'protective' of the patient's interests. These types of preferences may plausibly vary across decision-making contexts, but the research on this question has so far been almost exclusively performed in relation to medical AI. In this cross-sectional survey study we investigate the preferences of the adult Danish population for five specific protective features of AI systems and implementation across a range of eight different use cases in the public and commercial sectors ranging from medical diagnostics to the issuance of parking tickets. We find that all five features are seen as important across all eight contexts, but that they are deemed to be slightly less important when the implications of the decision made are less significant to the respondents.

## Introduction

Artificial intelligence systems based on deep learning architectures are being investigated as decision-support systems for human decision-makers across a wide range of decision-making contexts. In some cases such systems have already been implemented in practice and the processes of development and implementation are accelerating across many sectors of society [1–3].

It is known from the literature on AI in medicine that patients and the public hold relatively strong preferences in relation to desirable features of AI systems and their implementation, e.g. in relation to explainability and accuracy, and in relation to the role of the human

study design, data collection and analysis, decision to publish or preparation of the manuscript.

**Competing interests:** The authors have declared that no competing interests exist.

decision-maker in the decision chain [4–11]. A recent systematic review of empirical studies of AI ethics in medicine found that there were differences in views between patients and other stakeholders such as doctors and summarises the findings related to patients and relatives in the following way:

> "Overall, patients and family members expressed moderate support for medical AI, while identifying ethics as a major barrier/concern in accepting medical AI. Major ethical concerns included responsibility, privacy, data security, bias and accuracy, and lack of human interactions. Overall, patients and family members expressed concern that AI technologies would disengage physicians from the healthcare process, demonstrating a preference for physician involvement in diagnosis, decision-making, and clinical communication." [p 17, References removed] [12].

The features of AI systems and AI implementation that are identified in the literature as preferred by patients can be seen as 'protective' of the patient's interests in the sense that they allow the patient to know that AI has been used, that the performance is good, that the system is not discriminatory and that there is a proper role for a human professional in the decision-making chain.

These types of preferences may plausibly vary across decision-making contexts, but the research on this question has so far been almost exclusively performed in relation to medical AI.

In this study we investigate the preferences of the adult Danish population for specific protective features of AI systems and implementation across a range of different use cases in the public and commercial sectors.

The prior hypotheses are that there are 1) differences in preferences between different use cases depending on the perceived importance of the decision made, 2) differences in preferences in relation to the same use case depending on demographic factors, i.e. age, gender and level of education, and 3) differences in preferences depending on respondents' general attitudes towards AI, trust in human decision-makers, and self-assessed knowledge about AI.

We further hypothesise that these differences will mirror the differences in the literature on risk perception of new technologies, e.g. that older people will have stronger preferences for protective factors than younger people [13].

In order to investigate these hypotheses we chose a cross-sectional survey design in a representative sample of the adult Danish population.

## Materials and methods

An e-questionnaire was developed presenting eight different decision-making contexts, each involving a hypothetical decision about the respondent. These include medical diagnostics and seven other contexts that are all familiar to Danish citizens, either in their own lives or in the lives of relatives and friends. These are:

p Medical diagnostics
p Decision about early retirement pension
c Approval for consumer loan
p Police investigation of home burglary
c Determination of car insurance premium
p Ambulance dispatch
c Issuance of parking ticket
p Allocation of place in childrens' nursery

In Denmark five of these would be decisions made by public authorities or as part of the delivery of public services (p above), and three would be decisions made by commercial entities (c above).

As an introduction to the questionnaire the respondents were provided the following information about AI systems:

*Artificial Intelligence (AI) systems are used to make and advise on decisions in a wide range of areas. The AI system can either make the decisions itself or suggest a decision to a person who then makes the decisions. We are interested in knowing how acceptable you consider it to use AI systems to make decisions in various areas.*

This introduction was followed by a specific description of the choice situation for each of the eight decision-making contexts:

*In the following, you will be presented with a series of situations where an AI system makes a decision regarding you/your situation, when [context reference] is to be decided.*

Respondents were then asked to rate the importance of five protective features of an AI system used as decision support in each of the eight decision-making contexts. These five features were based on features identified in the literature on patient and population preferences in relation to medical AI. All items were of the form 'It is important for me that. . .' and the rating was on a 5 point Likert scale from 'Completely agree' (1) to 'Completely disagree' (5). The items were:

I have knowledge about AI involvement in the decision

A human being is responsible for the decision

I have knowledge that the AI system does not discriminate

A human being can explain the system decision

The system performs at least as well as a human decision-maker

For each context a simple summative scale was formed by addition of the five answers.

The respondents were also asked to rate their trust in human decision-makers in each context.

In order to minimise sequencing effects the eight decision-making contexts were presented in a random order to each respondent.

The questionnaire included scales measuring positive and negative general expectations about the societal effects of AI. Each scale contains four items rated on a 5 point Likert scale [14]. These scales have good psychometric reliability with Cronbach's Alpha of .869 for the positive scale and .821 for the negative scale.

The respondents were also asked to rate their own knowledge about AI on a 5 point Likert scale item from 'Nothing' to 'A lot'.

Respondents were sourced from Kantar's general panel which is representative of the adult Danish population. Potential participants were sent an invitation to participate by e-mail and one reminder e-mail in December 2021.

All members of the panel are adults and gave full informed consent to participation. Consent was collected in the e-survey system by respondents ticking a consent box prior to getting access to the survey itself.

Demographic information was collected about gender, age, geographical region, and level of education.

Statistical analysis was performed in IBM SPSS 29. The responses in relation to the importance of the features of the AI system are all heavily skewed to the left, and we have therefore

used non-parametric statistical methods, Mann Whitney U test for two group comparisons, Kruskal-Wallis test for comparisons between more than two groups, Friedmans ANOVA for one sample comparisons, and Spearman ordinal correlation [15]. We used Bonferroni correction to account for multiple significance tests and the p values reported are after Bonferroni correction, and therefore conservative.

This anonymous population survey did not require research ethics approval in Denmark, and therefore could not be submitted to the Danish research ethics committee system for approval.

## Results

The e-questionnaire was delivered to 900 members of the Kantar panel and 643 completed the questionnaire, response rate 71.4%. A sample efficiency analysis calculating the overall concordance between the respondents and the desired sample characteristics, considering obtained and desired numbers in relation to gender, age, geographical region, and level of education was performed giving a sample efficiency of 90.92%.

Of the respondents 323 (50.2%) are men and 320 women (49.8%). The age distribution is 18–35 years of age 136 (21.2%), 36–59 266 (41.4%), 60+ 241 (37.5%). 182 (28.3%) of the respondents have school education only, 284 (44.2%) have further education, and 177 (27.5%) higher education.

The respondents rate their own knowledge about AI as low. 208 (29.8%) state that they know 'Nothing' and 185 (26.5%) that they know 'A little'. The median score is 2 (= 'A little') and the average score 2.49 (SD 1.34).

The results concerning the importance of features of the AI system in each of the eight decision-making contexts are shown in Table 1. They show that although there are differences in the evaluation of the importance of these features across decision-making contexts, they are perceived as important in all contexts. The mean score is consistently below 2, i.e. more than 'Agree' to 'It is important for me that. . .' for all contexts, and the median score for most features is 1, i.e. 'Completely agree' indicating that more than 50% of respondents completely agree that all protective features are important across most contexts. The statistical analysis show that some of these small differences are statistically significant when compared to 'medical diagnosis' as the base case. There are also some statistically significant differences between men and women, and according to age and education (see Table 1).

The results for the two general expectation scales are shown in Table 2. They show that there are both positive and negative expectations to the future with AI among the respondents. The correlation between the two scales is as expected negative and statistically significant (rho = -.444, p<0.001).

For all of the contexts the score on the simple summative scale is positively correlated with the positive expectation scale and negatively correlated with the negative expectation scale, i.e. the features are seen as more important by those who have a general negative view of AI. The scores on all simple summative scales are also positively correlated with the item measuring trust in the human decision-maker in that context, i.e. the features are seen as more important as trust in the decision-maker declines. For three of the contexts—consumer loan, car insurance, and ambulance dispatch—the summative score is correlated positively with the rating of personal knowledge about AI, i.e. the features are seen as more important if knowledge is rated as low.

## Discussion

The response rate and the sample efficiency are both high and this indicates that the respondents are a reasonable approximation to the Danish adult population.

**Table 1. Importance of AI features across services/contexts.**

| AI system features | Knowledge about AI involvement in decision | Human being is responsible for decision | Knowledge that AI does not discriminate | Human being can explain system decision | The system performs at least as well as a human decision-maker | Trust in human decision-maker in this field |
|---|---|---|---|---|---|---|
| Field of application of AI | Mean (SD) Median (quartiles) Statistical tests | | | | | |
| Medical diagnostics | 1.61 (.92) 1 (1–2) g* a** | 1.36 (.65) 1 (1–2) a** | 1.53 (.84) 1 (1–2) g** a** | 1.49 (.78) 1 (1–2) g** a* | 1.57 (93) 1 (1–2) | 1.93 (.81) 1 (1–2) a** |
| Early retirement pension | 1.73 (1.02) 1 (1–2) a** | 1.61 (.88) 1 (1–2) a* C** | 1.58 (.84) 1 (1–2) a** | 1.59 (.80) 1 (1–2) a** | 1.69 (.98) 1 (1–2) | 2.37 (1.04) 2 (2–3) C** |
| Consumer loan | 1.85 (1.10) 1 (1–2) a** C** | 1.76 (1.00) 1 (1–2) a** C** | 1.62 (.84) 1 (1–2) a** | 1.69 (.90) 1 (1–2) g* a** C** | 1.73 (.94) 1 (1–2) g** | 2.15 (.90) 2 (2–3) C* |
| Police investigation | 1.76 (1.01) 1 (1–2) a** | 1.51 (.77) 1 (1–2) | 1.52 (.76) 1 (1–2) g* a** | 1.61 (.81) 1 (1–2) g* a** | 1.72 (.90) 1 (1–2) g** | 1.99 (.88) 2 (1–2) |
| Car insurance | 1.90 (1.06) 2 (1–3) a** C** | 1.79 (.96) 2 (1–2) a** C** | 1.60 (.82) 1 (1–2) g* a** | 1.72 (.90) 1 (1–2) g* a** C** | 1.68 (.87) 1 (1–2) g* | 2.18 (.88) 2 (2–3) C** |
| Ambulance dispatch | 1.88 (1.11) 1 (1–2) g* a** C** | 1.59 (.86) 1 (1–2) a** C** | 1.54 (.83) 1 (1–2) g* a** | 1.73 (.95) 1 (1–2) g** a** C** | 1.63 (.94) 1 (1–2) a** | 1.88 (.85) 1 (1–2) |
| Parking ticket | 1.87 (1.07) 2 (1–2) a** e** C** | 1.90 (1.08) 2 (1–3) C** | 1.68 (.90) 1 (1–2) a** | 1.78 (.98) 1 (1–2) a** C** | 1.80 (1.01) 1 (1–2) C* | 2.76 (1.23) 3 (2–4) g* e* C** |
| Place in childrens' nursery | 1.88 (.1.07) 2 (1–2.25) a** C** | 1.71 (.89) 1 (1–2) C** | 1.60 (.84) 1 (1–2) | 1.69 (.87) 1 (1–2) C** | 1.74 (.96) 1 (1–2) C* | 2.17 (.92) 2 (1–3) C** |
| Mean | 1,81 | 1,65 | 1,58 | 1,66 | 1,70 | 2,18 |

N = 643.

'Don't know' answers coded as missing. n(missing) 12–101.

g = gender–more important for females.

a = age–more important for older ages.

e = education- lowest and highest group differ from middle group.

C = comparison to 'Medical diagnostics' as base case.

* p < 0.05

** p < 0.01 (after Bonferroni correction).

Mann-Whitney U test for gender.

Kruskal-Wallis test for other background factors.

Friedman ANOVA test for comparisons to base case.

The respondents are asked to consider hypothetical situations which is a possible methodological weakness. The decision-making contexts are, however, all contexts of which the respondents will have either personal experience or experience from friends or relatives, and most have also been discussed extensively in the Danish media.

**Table 2. Expectation of general societal effects of AI.**

|  | Mean | SD |
|---|---|---|
| **Positive effects** |  |  |
| More jobs | 3.38 | 1.921 |
| Longer life | 3.58 | 1.966 |
| Better quality of life | 3.60 | 1.870 |
| Peace and political stability | 3.26 | 2.108 |
| Total | 13.82 | 6.66 |
| **Negative effects** |  |  |
| Unemployment | 3.51 | 1.649 |
| Unintended harm to human beings | 3.64 | 1.672 |
| Human loss of control to machines | 3.80 | 1.492 |
| Increased collection of data / mass surveillance | 4.41 | 1.152 |
| Total | 15.36 | 4.86 |

A possible weakness of the study is that we have not investigated the use of AI in contexts where the stakes in terms of the possible impact of a decision on personal interests are very low, or are perceived by the public to be very low. This may lead to an overestimation of the strength of the perceived importance of protective features. This is a plausible complaint, but primarily indicates that the scope of our conclusions is limited to contexts where something reasonably important is perceived to be at stake by the public. More research is needed on low risk / low stake, and perceived low risk / low stake contexts.

The results show that there are differences in the Danish general population in relation to how important different protective features of an AI system and the system use are perceived in different decision-making contexts. The original hypothesis is thus confirmed, but the differences found are generally small. The five features the respondents were asked about are seen as quite important in all contexts, but they are especially important when the stakes are high. They are rated as most important in relation to medical diagnostics, and least important in relation to the issuance of a parking ticket. There are no consistent differences between public and commercial contexts.

There is a general tendency that the importance of the features is rated higher by women, by older respondents, by persons at either end of the education spectrum, by those who rate their own knowledge of AI as low, and by those who have a general negative view of the future societal effects of AI. These findings also confirm the original hypotheses and are consistent with the general literature on risk perception [16–23]. The differences in stated importance of the features are all in the expected direction, except in relation to education where the results found are difficult to interpret.

Denmark is in general a 'high trust' society, where citizens trust each other and trust government institutions [24–26]. We found that the protective features were rated as more important when trust in the human decision-maker in a particular context was lower. This finding may be important when considering the implications of our findings in 'low trust' societies.

## Conclusions

The results strongly indicate that the general population wants AI systems that are used to support important decisions about them in the public or the commercial sector to protect their interests in relation to a wide range of protective factors. They want the systems to perform as well as a human decision-maker, to be explainable, and to be non-discriminatory. In addition, they want to know that AI has been used, and they want the human decision-maker to make

the final decision. These findings further point towards these features as being important design parameters in the development of AI systems and in their implementation in decision chains.

Although there are statistically significant differences in preferences according to factors such as age or self-assessed knowledge about AI these differences are small. The old and those who do not know much about AI have stronger preferences for 'protective' features than the young and knowledgeable, but even the young and knowledgeable strongly prefer such features. Developers and regulators can therefore not expect these preferences to change due to generational change or increases in knowledge.

The results also have potential policy implications for the regulation of AI use. The most important policy implication is probably that the general public has similar views on the importance of 'protective' features of AI systems and AI implementation across sectors, making no real distinction between public and private sectors and activities. This may be an argument against sector specific regulation. However, the design of the study does not make it possible to study trade-offs between different protective features within a given context, or differential willingness to pay for a specified level of protection between contexts. Further research using discrete choice methods is necessary to provide more fine-grained information to guide policy makers.

## Author Contributions

**Conceptualization:** Søren Holm, Thomas Ploug.

**Data curation:** Søren Holm.

**Formal analysis:** Søren Holm.

**Funding acquisition:** Thomas Ploug.

**Investigation:** Søren Holm, Thomas Ploug.

**Methodology:** Søren Holm, Thomas Ploug.

**Project administration:** Thomas Ploug.

**Writing – original draft:** Søren Holm, Thomas Ploug.

**Writing – review & editing:** Søren Holm, Thomas Ploug.

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
