## [Decision Letter · Decision Letter 0]

11 Jul 2023

PONE-D-23-18987Population preferences for AI system features across eight different decision-making contextsPLOS ONE

Dear Dr. Holm,

Thank you for submitting your manuscript to PLOS ONE. After careful consideration, we feel that it has merit but does not fully meet PLOS ONE’s publication criteria as it currently stands. Therefore, we invite you to submit a revised version of the manuscript that addresses the points raised during the review process.

We look forward to receiving your revised manuscript.

Kind regards,

Radoslaw Wolniak, full professor

Academic Editor

PLOS ONE

Journal Requirements:

2. a. For studies reporting research involving human participants, PLOS ONE requires authors to confirm that this specific study was reviewed and approved by an institutional review board (ethics committee) before the study began. Please provide the specific name of the ethics committee/IRB that approved your study, or explain why you did not seek approval in this case.

b. Please provide additional details regarding participant consent. In the ethics statement in the Methods and online submission information, please ensure that you have specified what type you obtained (for instance, written or verbal, and if verbal, how it was documented and witnessed). If your study included minors, state whether you obtained consent from parents or guardians. If the need for consent was waived by the ethics committee, please include this information.

Additional Editor Comments:

Please adjust the paper accoring to reviewers comments.. 

Reviewers' comments:

Reviewer's Responses to Questions

**Comments to the Author**

1. Is the manuscript technically sound, and do the data support the conclusions?

Reviewer #1: Yes

Reviewer #2: Yes

Reviewer #3: Partly

2. Has the statistical analysis been performed appropriately and rigorously? 

Reviewer #1: Yes

Reviewer #2: Yes

Reviewer #3: Yes

3. Have the authors made all data underlying the findings in their manuscript fully available?

Reviewer #1: Yes

Reviewer #2: Yes

Reviewer #3: Yes

4. Is the manuscript presented in an intelligible fashion and written in standard English?

Reviewer #1: Yes

Reviewer #2: Yes

Reviewer #3: No

5. Review Comments to the Author

Reviewer #1: Dear Authors,

Congratulations for your interesting research.

The title and the abstract coincide with the content of the paper. Keywords are well-chosen.

• The problem and goal of the paper are clearly and correctly formulated.

• The manuscript is clear, relevant for the field and presented in a well-structured manner.

• To achieve the goal of the paper, the applied research methods can be considered correct.

• The paper contains a comprehensive reference list. The literature studies presented should be considered sufficient both as to the proper selection of sources and their quantity. The cited references are current.

• The manuscript’s results are reproducible based on the details given in the methods section.

• The figures and the tables are appropriate. They properly show the data. They are easy to interpret and understand.

I have some suggestions on how to make your text more attractive

The main problem is the epistemological structure (why the article was conceived and how the study was developed). I suggest the following structure of objectives: (i) research gap; (ii) research question; (iii) purpose of the article; (iv) intermediate objectives; (v) assumptions or hypo; and (vi) research method. This structure must appear in the introduction.

In conclusion, I propose

-evaluate the critical research, show its limitations and weaknesses,

- highlight the new knowledge and the lessons learned from it,

- describe the importance of the research and how it affects the wider field, show how the information obtained can be further used

Conclusions must be clearly and unambiguously linked to the results of the survey. Their theoretical and practical implications should be indicated.

Reviewer #2: The article deals with a very interesting topic. I propose to some extent to extend the literature review as an introduction to the empirical part of the article. The analysis of the empirical results is correct. The discussion part of the article should to some extent refer to other research conducted in this field.

Reviewer #3: The article concerns an interesting research topic. However, I have a few objections to that study and I would like to ask the authors to refer to the following issues:

1. Lack of a clear specification for what purpose the research was conducted. 2. No research purpose in the abstract. 3. The information about AI given in the introduction is insufficient. The introduction to AI applications in medical research should be expanded.

4. Please specify what type of artificial intelligence was used?

5. Please clearly and accurately describe the methodology used

6. The discussion needs improvement. In its current form, it is only an interpretation of the results, but without comparison with other research conducted by other scientists. 7. There is no summary in the work

8. Poor literature - 13 items. It needs to be greatly enriched.

6. PLOS authors have the option to publish the peer review history of their article (what does this mean?). If published, this will include your full peer review and any attached files.

Reviewer #1: No

Reviewer #2: No

Reviewer #3: No

---

## [Author Response · Author response to Decision Letter 0]

28 Sep 2023

https://journals.plos.org/plosone/s/file?id=wjVg/PLOSOne_formatting_sample_main_body.pdf [journals.plos.org] and 

https://journals.plos.org/plosone/s/file?id=ba62/PLOSOne_formatting_sample_title_authors_affiliations.pdf [journals.plos.org]

2. a. For studies reporting research involving human participants, PLOS ONE requires authors to confirm that this specific study was reviewed and approved by an institutional review board (ethics committee) before the study began. Please provide the specific name of the ethics committee/IRB that approved your study, or explain why you did not seek approval in this case.

Surveys of adults do not require research ethics approval in Denmark, and there are therefore no committees to which such projects can be submitted for approval. 

For additional information about PLOS ONE ethical requirements for human subjects research, please refer to http://journals.plos.org/plosone/s/submission-guidelines#loc-human-subjects-research. [journals.plos.org]

b. Please provide additional details regarding participant consent. In the ethics statement in the Methods and online submission information, please ensure that you have specified what type you obtained (for instance, written or verbal, and if verbal, how it was documented and witnessed). If your study included minors, state whether you obtained consent from parents or guardians. If the need for consent was waived by the ethics committee, please include this information.

We have added more details about consent in the Methods

Done

For additional information about PLOS ONE ethical requirements for human subjects research, please refer to http://journals.plos.org/plosone/s/submission-guidelines#loc-human-subjects-research [journals.plos.org].

The data are already available in the University of Manchester’s Figshare repository and the DOI is provided.

Additional Editor Comments:

Please adjust the paper accoring to reviewers comments.. 

Reviewers' comments:

Reviewer's Responses to Questions

Comments to the Author

1. Is the manuscript technically sound, and do the data support the conclusions?

Reviewer #1: Yes

Reviewer #2: Yes

Reviewer #3: Partly

2. Has the statistical analysis been performed appropriately and rigorously? 

Reviewer #1: Yes

Reviewer #2: Yes

Reviewer #3: Yes

3. Have the authors made all data underlying the findings in their manuscript fully available?

The PLOS Data policy [plosone.org] requires authors to make all data underlying the findings described in their manuscript fully available without restriction, with rare exception (please refer to the Data Availability Statement in the manuscript PDF file). The data should be provided as part of the manuscript or its supporting information, or deposited to a public repository. For example, in addition to summary statistics, the data points behind means, medians and variance measures should be available. If there are restrictions on publicly sharing data—e.g. participant privacy or use of data from a third party—those must be specified.

Reviewer #1: Yes

Reviewer #2: Yes

Reviewer #3: Yes

4. Is the manuscript presented in an intelligible fashion and written in standard English?

Reviewer #1: Yes

Reviewer #2: Yes

Reviewer #3: No

5. Review Comments to the Author

Reviewer #1: Dear Authors,

Congratulations for your interesting research.

Thank You!

The title and the abstract coincide with the content of the paper. Keywords are well-chosen.

• The problem and goal of the paper are clearly and correctly formulated.

• The manuscript is clear, relevant for the field and presented in a well-structured manner.

• To achieve the goal of the paper, the applied research methods can be considered correct.

• The paper contains a comprehensive reference list. The literature studies presented should be considered sufficient both as to the proper selection of sources and their quantity. The cited references are current.

• The manuscript’s results are reproducible based on the details given in the methods section.

• The figures and the tables are appropriate. They properly show the data. They are easy to interpret and understand.

I have some suggestions on how to make your text more attractive

The main problem is the epistemological structure (why the article was conceived and how the study was developed). I suggest the following structure of objectives: (i) research gap; (ii) research question; (iii) purpose of the article; (iv) intermediate objectives; (v) assumptions or hypo; and (vi) research method. This structure must appear in the introduction.

We have comprehenseively rewritten the Introduction to take account of these helpful comments on structure.

In conclusion, I propose

-evaluate the critical research, show its limitations and weaknesses,

- highlight the new knowledge and the lessons learned from it,

- describe the importance of the research and how it affects the wider field, show how the information obtained can be further used

Conclusions must be clearly and unambiguously linked to the results of the survey. Their theoretical and practical implications should be indicated.

We have introduced a Conclusion section taking into account these proposed changes and especially discussing practical and regulatory implications.

Reviewer #2: The article deals with a very interesting topic. I propose to some extent to extend the literature review as an introduction to the empirical part of the article. The analysis of the empirical results is correct. The discussion part of the article should to some extent refer to other research conducted in this field.

We have made changes to the Introduction and Conclusion to take into account the very similar comments of Reviewer 1 & 2

Reviewer #3: The article concerns an interesting research topic. However, I have a few objections to that study and I would like to ask the authors to refer to the following issues:

1. Lack of a clear specification for what purpose the research was conducted. 2. No research purpose in the abstract. 3. The information about AI given in the introduction is insufficient. The introduction to AI applications in medical research should be expanded.

We have comprehensively rewritten the Introduction.

4. Please specify what type of artificial intelligence was used?

The study does not use any AI. It asks respondents to reflect on the use of (any kind of) AI in particular use cases.

5. Please clearly and accurately describe the methodology used

We have gone through the description of the methodology and tried to clarify any issue we could identify.

6. The discussion needs improvement. In its current form, it is only an interpretation of the results, but without comparison with other research conducted by other scientists. 7. There is no summary in the work

We have added a summary in the Conclusion and revised it in accordance with the comments of all three reviewers.

8. Poor literature - 13 items. It needs to be greatly enriched.

We have provided a larger review of the literature in the Introduction, and there are now 26 references.

---

## [Decision Letter · Decision Letter 1]

20 Nov 2023

Population preferences for AI system features across eight different decision-making contexts

PONE-D-23-18987R1

Dear Dr. Holm,

We’re pleased to inform you that your manuscript has been judged scientifically suitable for publication and will be formally accepted for publication once it meets all outstanding technical requirements.

Kind regards,

Maurizio Fiaschetti

Academic Editor

PLOS ONE

Additional Editor Comments (optional):

Reviewers' comments:

Reviewer's Responses to Questions

**Comments to the Author**

1. If the authors have adequately addressed your comments raised in a previous round of review and you feel that this manuscript is now acceptable for publication, you may indicate that here to bypass the “Comments to the Author” section, enter your conflict of interest statement in the “Confidential to Editor” section, and submit your "Accept" recommendation.

Reviewer #1: (No Response)

Reviewer #2: (No Response)

Reviewer #3: All comments have been addressed

2. Is the manuscript technically sound, and do the data support the conclusions?

Reviewer #1: (No Response)

Reviewer #2: Yes

Reviewer #3: Yes

3. Has the statistical analysis been performed appropriately and rigorously? 

Reviewer #1: (No Response)

Reviewer #2: Yes

Reviewer #3: Yes

4. Have the authors made all data underlying the findings in their manuscript fully available?

Reviewer #1: (No Response)

Reviewer #2: Yes

Reviewer #3: Yes

5. Is the manuscript presented in an intelligible fashion and written in standard English?

Reviewer #1: (No Response)

Reviewer #2: Yes

Reviewer #3: Yes

6. Review Comments to the Author

Reviewer #1: The authors have adequately addressed the comments made in the previous round of reviews. I recommend publishing the article in its current version.

Reviewer #2: In the concluding section of the article, the discussion should be carried out with more reference to the literature. In conclusion, the recommendations made still need to be made more specific and concrete.

Reviewer #3: (No Response)

7. PLOS authors have the option to publish the peer review history of their article (what does this mean?). If published, this will include your full peer review and any attached files.

Reviewer #1: No

Reviewer #2: No

Reviewer #3: No

---

## [Editor Report · Acceptance letter]

24 Nov 2023

PONE-D-23-18987R1 

Population preferences for AI system features across eight different decision-making contexts 

Dear Dr. Holm:

I'm pleased to inform you that your manuscript has been deemed suitable for publication in PLOS ONE. Congratulations! Your manuscript is now with our production department. 

Kind regards, 

on behalf of

Dr. Maurizio Fiaschetti 

Academic Editor

PLOS ONE